taxonomy and systematics

Neomeniomorpha, Solenogastres, Chaetodermomorpha, Caudofoveata, *Apodomenia*

**Author for correspondence:**
Kevin M. Kocot
e-mail: kmkocot@ua.edu

# Phylogenomics of Aplacophora (Mollusca, Aculifera) and a solenogaster without a foot

Kevin M. Kocot[1], Christiane Todt[2], Nina T. Mikkelsen[3] and Kenneth M. Halanych[4]

[1]The University of Alabama and the Alabama Museum of Natural History, 500 Hackberry Lane, Tuscaloosa, AL 35487, USA
[2]Rådgivende Biologer AS, Edvard Griegs vei 3, 5059 Bergen, Norway
[3]University Museum of Bergen, The Natural History Collections, University of Bergen, Allégaten 41, 5007 Bergen, Norway
[4]Department of Biological Sciences, Auburn University, Auburn, AL 36849, USA

(iD) KMK, 0000-0002-8673-2688

Recent molecular phylogenetic investigations strongly supported the placement of the shell-less, worm-shaped aplacophoran molluscs (Solenogastres and Caudofoveata) and chitons (Polyplacophora) in a clade called Aculifera, which is the sister taxon of all other molluscs. Thus, understanding the evolutionary history of aculiferan molluscs is important for understanding early molluscan evolution. In particular, fundamental questions about evolutionary relationships within Aplacophora have long been unanswered. Here, we supplemented the paucity of available data with transcriptomes from 25 aculiferans and conducted phylogenomic analyses on datasets with up to 525 genes and 75 914 amino acid positions. Our results indicate that aplacophoran taxonomy requires revision as several traditionally recognized groups are non-monophyletic. Most notably, Cavibelonia, the solenogaster taxon defined by hollow sclerites, is polyphyletic, suggesting parallel evolution of hollow sclerites in multiple lineages. Moreover, we describe *Apodomenia enigmatica* sp. nov., a bizarre new species that appears to be a morphological intermediate between Solenogastres and Caudofoveata. This animal is not a missing link, however; molecular and morphological studies show that it is a derived solenogaster that lacks a foot, mantle cavity and radula. Taken together, these results shed light on the evolutionary history of Aplacophora and reveal a surprising degree of morphological plasticity within the group.

## 1. Introduction

The two groups of worm-like aplacophoran molluscs, Solenogastres (= Neomeniomorpha) and Caudofoveata (=Chaetodermomorpha), have perplexed biologists since their discovery [1,2]. Aplacophorans are characterized by a narrow or completely reduced foot, a unique posterior dorsoterminal sensory organ, and a small mantle cavity restricted to the posterior-most part of the body. Solenogasters and caudofoveates both completely lack a shell, but instead are covered in a dense coat of spiny or scale-like calcareous sclerites [3–8].

Aplacophorans have generally been regarded as early-branching molluscs and therefore have been central to questions surrounding the origin and early evolution of the phylum. Whether Solenogastres and Caudofoveata constitute a monophyletic taxon, Aplacophora [4,9,10], or a 'basal', paraphyletic grade [5,8,11–14], has been debated [6,7,15]. Recent molecular studies [16–18] have strongly supported monophyly of Aplacophora and a sister group relationship of Aplacophora and Polyplacophora (chitons), consistent with the Aculifera hypothesis [4]. Analyses of fossils (e.g. [19]) and evolutionary developmental

approaches [20,21] have provided further evidence for this hypothesis. Support for Aculifera has had an important impact on understanding of plesiomorphic characteristics of Mollusca [16] as it suggests the last common ancestor of the phylum was a large-bodied, chiton-like animal, but many more questions remain unanswered. Although aplacophorans are not the sister taxon to all other molluscs as previously thought [8,13,14,22,23], resolving aplacophoran phylogeny is critical to understanding early molluscan evolution, as it could help reveal the evolutionary polarity of key morphological characters for Aplacophora, Aculifera and Mollusca as a whole.

Caudofoveate taxonomy is based primarily on characteristics of the sclerites and radula. Around 130 species have been described and three families are traditionally recognized [6,24]. Limifossoridae has been hypothesized to show the most plesiomorphic morphological characters among caudofoveates [25–27], mainly a solenogaster-like (distichous) radula with two teeth per row and a simple body shape. Within the more diverse Solenogastres, classification is based primarily on characters of the sclerites, cuticle, radula, ventrolateral foregut glands and reproductive anatomy [24,28,29]. Presently, around 280 species in 24 families and four orders are recognized, but the actual diversity within the group is estimated to be considerably higher [6]. According to the taxonomy established by Salvini-Plawen [28], the orders Pholidoskepia and Neomeniamorpha are grouped together in a higher taxon called Aplotegmentaria. The small-bodied, scale-bearing Pholidoskepia have been regarded as 'primitive' solenogasters [28,30,31]. The remaining two orders, Cavibelonia and Sterrofustia, are grouped together in a higher taxon called Pachytegmentaria.

Chitons have a fairly rich fossil record [32] and their phylogeny is at least generally understood [33–36]. However, no bona fide solenogaster or caudofoveate fossils are known [19,37–40], and cladistic morphological analyses examining solenogaster phylogeny [5,41,42] have generally failed to recover most higher-level taxa monophyletic, suggesting that the existing taxonomy does not reflect the evolutionary history of the group or that the morphological data analysed lack sufficient phylogenetic signal to reconstruct aplacophoran relationships. Recent molecular studies employing datasets dominated by nuclear ribosomal and mitochondrial genes [43–45] have greatly improved understanding of relationships within Caudofoveata. However, nuclear ribosomal genes are GC rich in Solenogastres [46,47] and universal primers for mitochondrial genes do not work well on some aplacophorans [45]. Here, we employed a phylogenomic approach to reconstruct a broad-scale phylogenetic framework for Aplacophora. In the light of the reconstructed phylogenetic framework, including a newly described and highly unusual lineage, we assessed the monophyly of traditionally recognized aplacophoran taxa and implications for understanding early molluscan evolution.

## 2. Material and methods

### (a) Taxon sampling and morphological work

We aimed to sample transcriptome data from as many recognized aplacophoran taxa and as broad a range of morphological disparity as possible (electronic supplementary material, tables S1 and S2). The identification of specimens and data collection for description of Apodomenia enigmatica sp. nov. involved examination of sclerites, radulae (if present) and internal anatomy following standard approaches of [48,49]. Scanning electron microscopy was conducted on dried, uncoated specimens using a Phenom Pro with an accelerating voltage of 5 kV. When possible, voucher specimens of species sampled herein were deposited into the University Museum of Bergen or the Alabama Museum of Natural History (see below).

### (b) Molecular techniques

Because prey nucleic acid contamination in solenogasters has been problematic in previous molecular studies [46,47], specimens were starved in the laboratory prior to preservation whenever possible (electronic supplementary material, table S1). Specimens of all taxa were preserved in RNAlater and stored at $-80°C$ or frozen at $-80°C$. Different RNA extraction approaches were employed depending on sample size (electronic supplementary material, table S1). Total RNA concentration and purity were estimated using a NanoDrop 2000 (Thermo Scientific) and RNA quality was evaluated on a 1% SB agarose gel. For most taxa, cDNA library preparation and sequencing was performed as described in [50]. For Chaetoderma nitidulum, Falcidens sagittiferus, Stylomenia sulcodoryata and Tonicella lineata, total RNA was sent to Macrogen (South Korea) for Illumina stranded library preparation and sequencing using 1/4 lane of an Illumina HiSeq 2500 with $2 \times 100$ bp paired-end sequencing.

### (c) Dataset assembly

For most taxa, digital normalization and assembly were performed as described in [50]. For taxa sequenced at Macrogen and publicly available Rhyssoplax and Pholidoskepia sp. (misidentified as Chaetoderma sp. by Zapata et al. [51]; see [52]), read trimming, digital normalization and assembly were performed using the 3/2014 version of TRINITY. Contigs from all taxa were translated with TRANSDECODER and translated sequences shorter than 50 amino acids (AAs) were deleted.

For orthology inference, we employed HAMSTR 13 [53], using a custom core orthologue set based on transcriptome data from Alexandromenia crassa, A. enigmatica, Helluoherpia aegiri, Leptochiton sp., Neomenia carinata, Prochaetoderma californicum, Simrothiella margaritacea and the genome of Lottia gigantea following [50]. In cases where one of the first or last 20 characters of an AA sequence was an X, all characters between the X and that end of the sequence were deleted and treated as missing data. Each gene was then aligned with MAFFT [54] and alignments were trimmed with ALISCORE [55] and ALICUT [56] to remove ambiguously aligned regions. A consensus sequence was inferred for each alignment using infoalign [57] and the percentage of positions of a sequence that differed from the consensus of the alignment were calculated using the infoalign's 'change' calculation. Any sequence with a value greater than 75 was deleted. Sequence regions containing lesser than or equal to 20 AAs in length surrounded by 10 or more gaps on either side were deleted. We deleted sequences that did not overlap with all other sequences in the alignment by greater than or equal to 20 AAs, starting with the shortest sequence.

In some cases, a taxon was represented in an alignment by two or more sequences. We built trees in FASTTREE 2 [58] using the 'slow' option and used PHYLOTREEPRUNER [59] to select the best sequence for each taxon. Only genes sampled for 20+ taxa after pruning with PhyloTreePruner were retained. To further screen for paralogy and contamination, we used TRESPEX [60] to search for gene trees where select, well-established monophyletic groups (Conchifera, Polyplacophora, Pholidoskepia, Amphimeniidae, Neomeniidae and Prochaetodermatidae) were recovered non-monophyletic with strong support (bootstrap support greater than 95) and excluded those 12 genes from further

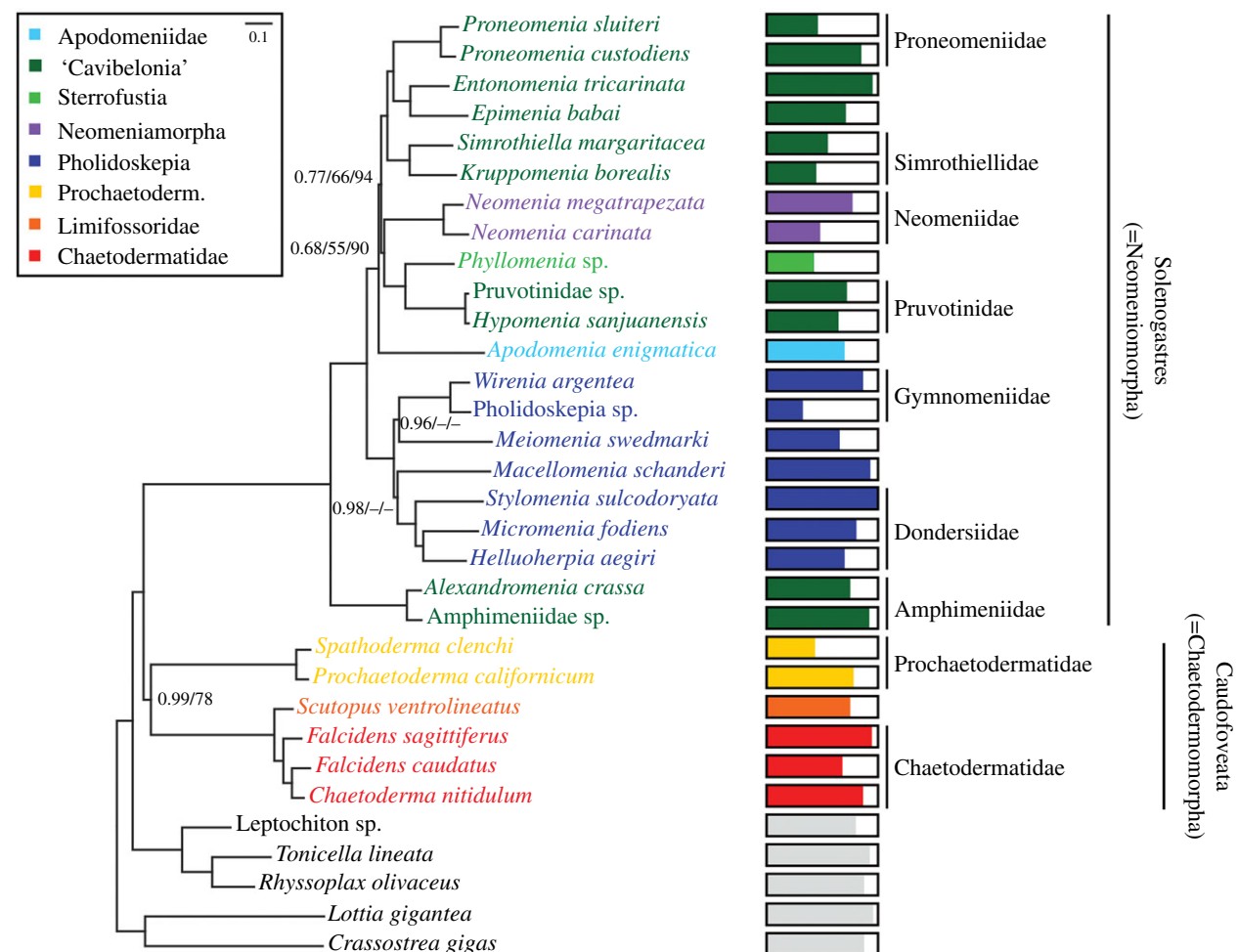

**Figure 1.** Phylogeny of Aplacophora based on 200 best genes in terms of branch-length heterogeneity. BI topology shown with posterior probabilities/RAxML/IQ-TREE bootstrap support values less than 1.0/100/100 shown at each node (see electronic supplementary material, figures S2 – S7 for ML topologies). Coloured bars show the proportion of genes sampled for each taxon. (Online version in colour.)

consideration. This yielded a complete data matrix with 525 genes that was 75 914 AAs long with 30.43% missing data (electronic supplementary material, figure S1A). We also measured branch-length heterogeneity (LB) score as calculated in TRESPEX to identify genes most likely to be susceptible to long-branch attraction and assembled a data matrix with the best 200 genes according to LB, which was 30 185 AAs long with 30.7% missing data (electronic supplementary material, figure S1B).

### (d) Phylogenetic analyses

Phylogenetic analyses were conducted for all data matrices using maximum likelihood (ML) in RAxML 7.3.8 [61] with the '-f a' flag, which specifies a search for best-scoring ML tree and a rapid bootstrap analysis in one program run. Each matrix was partitioned by gene and analysed with the PROTGAMMALGF model. Nodal support was assessed with 100 rapid bootstraps (-N 100).

ML analyses were also performed on all matrices in IQ-TREE [62] using the site-heterogeneous PMSF model [63] (-m LG + C60 + G + F) with the RAxML bipartitions tree provided as the required guide tree (-ft). Nodal support was assessed with 1000 rapid bootstraps (-bb 1000).

Bayesian inference (BI) analysis was conducted in PHYLOBAYES MPI 1.2f [64] with the site-heterogeneous CAT-GTR-G4 model. Because of the computational intensity of BI, only the matrix of the 200 least branch-length heterogeneous genes was analysed using this method. Four parallel chains were run for approximately 8000 cycles each with the first 2000 trees discarded as burn-in. A 50% majority rule consensus tree was computed from the remaining trees from each chain. PHYLOBAYES bpcomp

maxdiff of 0.1565 and meandiff of 0.0046 indicated that all chains had converged.

### (e) DNA barcoding

We sequenced cytochrome *c* oxidase subunit I (COI) from specimens of *A. enigmatica* sp. nov. spanning its known geographical range following the laboratory approaches of [45] or by transcriptome sequencing as described above. ML-corrected substitutions per site were calculated in MEGA 7 using the maximum composite likelihood parameter with a γ parameter of 1.0 [65].

## 3. Results

### (a) Phylogenetic analyses

Because aplacophorans have exhibited relatively long branches in previous phylogenomic studies [16,17] and *Falcidens caudatus* was on an extremely long branch in the ML analysis of all 525 genes (electronic supplementary material, figures S2 and S3), we conducted analyses of all 525 genes excluding *F. caudatus* (electronic supplementary material, figures S4 and S5) and sorted genes by LB as calculated in TRESPEX [60] and assembled and analysed a reduced dataset of just the 200 genes with the lowest branch-length heterogeneity (figure 1; electronic supplementary material, figures S6 and S7 and tables S3 and S4). Because analyses of the complete dataset (electronic supplementary material, figures S2 and S3) yielded similar results as that of the

reduced dataset, we focus our discussion on analyses of the reduced dataset and highlight notable differences when applicable. Details on data matrices analysed are presented in electronic supplementary material, tables S3 and S4.

Analyses of the dataset with reduced LB strongly supported Polyplacophora (BI posterior probability [pp]/RAxML bootstrap support [bs]/IQ-TREE bs = 1.00/100/100), Aplacophora (1.00/86/100), Solenogastres (1.00/100/100) and Caudofoveata (0.99/78/100). Within Solenogastres, Cavibelonia is polyphyletic. Amphimeniidae was recovered as the sister taxon of all other sampled lineages of Solenogastres with maximal support. The remaining cavibelonians along with the one sampled representative of Sterrofustia (*Phyllomenia* sp.), Neomeniamorpha and *A. enigmatica* sp. nov. formed a maximally supported clade, which was recovered as the sister taxon of Pholidoskepia. Within this clade, *Phyllomenia* formed a clade with Pruvotinidae with maximal support. The clade of Neomeniamorpha, *Phyllomenia* and Pruvotinidae was recovered as the sister group to a well-supported clade consisting of the remaining 'cavibelonian' taxa: *Epimenia, Entonomenia*, Proneomeniidae and Simrothiellidae; support for placement of this clade was also variable (0.77/66/94).

We recovered Pholidoskepia monophyletic with full support in all analyses. Dondersiidae was recovered with maximal support in all analyses. However, relationships among families differed among analyses. In the BI analysis, Macellomeniidae was recovered sister to Dondersiidae with relatively strong support (pp = 0.98). Macellomeniidae was recovered sister to Gymnomeniidae in the ML analyses, but with weak support (electronic supplementary material, figures S6 and S7). Meiomeniidae was recovered in a clade with Gymnomeniidae with moderate support in BI (pp = 0.96) but as the sister taxon of all other pholidoskepians in ML with moderate to weak support (electronic supplementary material, figures S6 and S7).

Within Caudofoveata, we sampled at least one member of each recognized family and recovered a well-supported Chaetodermatidae (*Falcidens* + *Chaetoderma*) with maximal support (1.00/100). *Chaetoderma* was nested within *Falcidens* with *C. nitidulum* and *Falcidens caudatus* forming a clade with maximal support.

## (b) *Apodomenia enigmatica* sp. nov

Distinguishing between the two major lineages of Aplacophora is generally straightforward: caudofoveates lack a foot but have an anterior muscular structure called the oral shield, whereas solenogasters have a narrow, midventral foot and lack an oral shield. However, during two recent Antarctic research expeditions, specimens of an aplacophoran, which lacks both a foot and an oral shield, were found inside *Rossella* sp. sponges (electronic supplementary material, table S5). We sequenced COI from six specimens (GenBank MK404651–MK404656) spanning the known geographical range of the species. Only 15 of 625 nucleotide positions in the amplified region were variable (ML-corrected substitutions per site = 0.008; electronic supplementary material, figure S8), suggesting that all of the sampled specimens belong to the same species.

Apodomeniidae fam. nov.

Diagnosis: cuticle thick, sclerites acicular and in one layer; foot reduced; radula and ventrolateral foregut glands lacking; spawning duct with extraepithelial gland cells.

*Apodomenia* gen. nov.

Diagnosis: sclerites solid acicular spines; radula and ventrolateral foregut glands lacking; foot lacking, foot groove covered by the cuticle and sclerites; common vestibulo-buccal opening; secondary genital opening unpaired; mantle cavity reduced.

Etymology: '*Apodo*' from *apodus* (lat.) 'lacking a foot'; '*menia*' is a common suffix for solenogaster genus names that is derived from '-*mene*' (gr.) referring to the moon or crescent.

*Apodomenia enigmatica* sp. nov.

Type species for *Apodomenia* gen. nov., by monotypy.

Diagnosis: Body up to about 16 cm long, slender and very stiff. Ventral groove and foot lacking. Cuticle thick, with robust mantle sclerites arranged in a right angle to body surface. Sclerites are flattened solid spines. Vestibulum with a few simple sensory papillae. Mouth opening within vestibulum. Radula and ventrolateral foregut glands lacking. Midgut with paired anteriodorsal caecum; without regular constrictions. Spawning ducts partly fused, ciliated and surrounded by extraepithelial gland cells, the cell bodies of which lie distally to a thick muscular coat. One pair of branched seminal vesicles. Mantle cavity highly reduced, lacking respiratory folds.

Type material: Holotype (ZMBN 129503): two histological section series (anterior + posterior). Paratype 1 (ZMBN 129501): one histological section series (anterior). Paratype 2 (ZMBN 129505): large specimen incomplete at posterior end, fixed in 4% formalin and preserved in 70% ethanol. Paratype 3 (ZMBN 129502): posterior end broken, anterior end dissected, fixed in 4% formalin and preserved in 70% ethanol. Paratype 4 (ALMNH 21269): one complete specimen broken at midbody, fixed in 4% formalin, stained with phosphomolybdic acid and preserved in 70% ethanol. Paratype 5 (ALMNH 21270): stained with phosphomolybdic acid, and preserved in 95% ethanol. Sample data for all specimens collected are presented in electronic supplementary material, table S5. Holotype and paratypes 1–3 are deposited in the University Museum of Bergen (ZMBN) and paratypes 5–6 are deposited in the Alabama Museum of Natural History (ALMNH).

Type locality: Wright's Gulf, Antarctica (73°17.7997 S, 129°11.5466 W) at 506 m in association with *Rossella* sp. Collected 25 January 2013.

Etymology: '*enigmatica*', from lat. *enigmaticus, -a, -um*, meaning mysterious, refers to the highly unusual morphology and lifestyle of the species.

Description: animals uniformly cylindrical-elongate, tending to curl up spirally when disturbed and during fixation. Largest specimen found (paratype 4; figure 2*a*) 155 mm long, with a maximum diameter of 8 mm; tip of posterior end missing. Animals completely covered in thick cuticle pierced by evenly sized massive, flattened spines. Spines arranged at a right angle to the body surface, resulting in an overall velvety appearance. Cuticle translucent and thus, on a closer look, body surface appears rather spiny even though only the tips of the spines protrude from the cuticle (figure 2*b,c*). In living animals, yellowish organs (gonad and midgut) and red hemolymph visible through the integument. Ventral furrow lacking, but sometimes the ventral side close to the anterior end appears slightly flattened. Areas with thin cuticle and distinctly smaller spines surrounding mouth and area around the anus and genital opening (figure 2*d,e*).

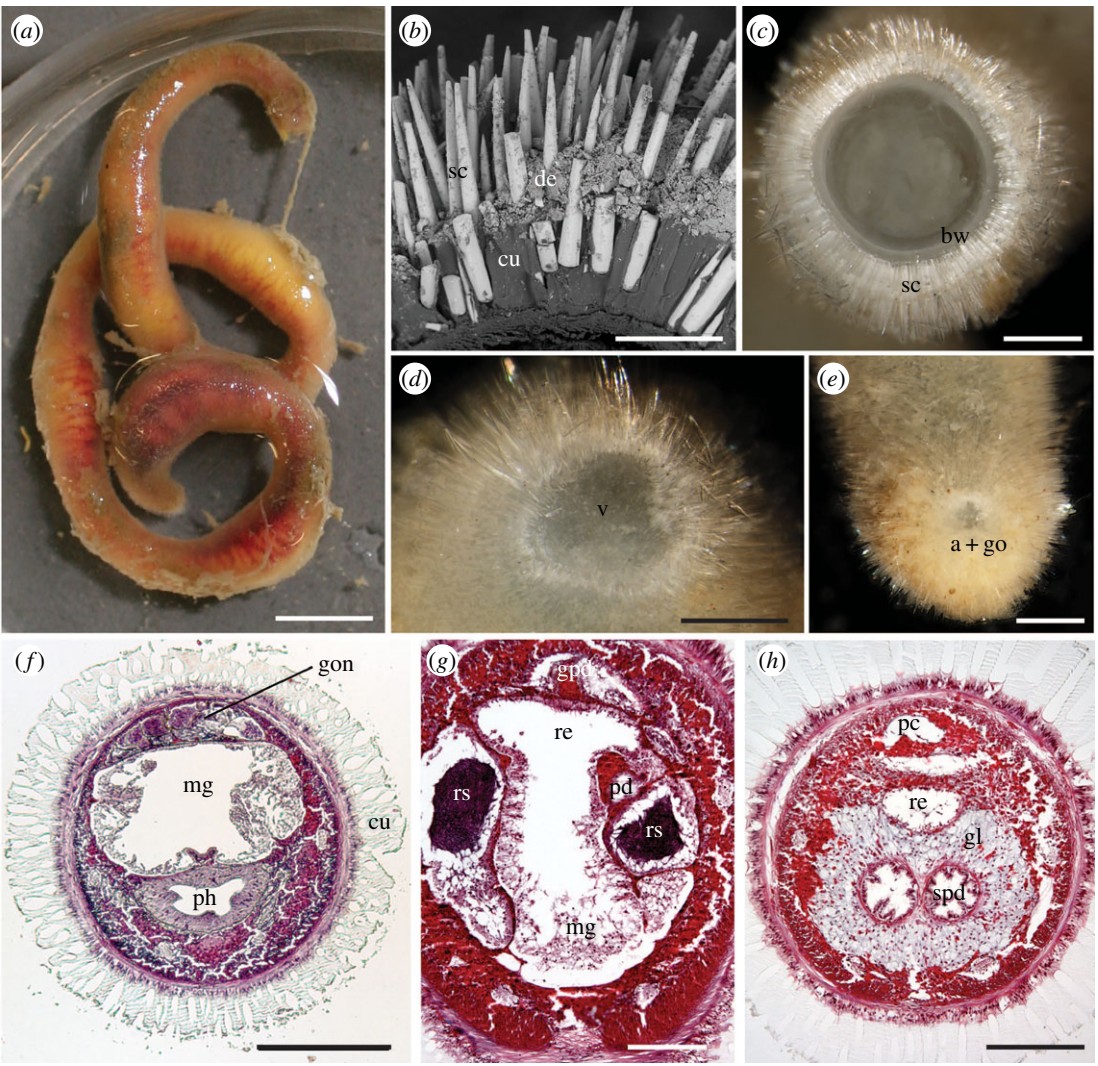

**Figure 2.** *Apodomenia enigmatica* sp. nov. (*a*) Habitus with (broken) posterior end above ( paratype 4). Scale bar, 8 mm. (*b*) Scanning electron micrograph of section through the midbody of ALMNH 21271. Scale bar, 200 μm. (*c*–*e*) Holotype, light microscopic images. (*c*) Cross-section through the body wall in the midgut region. Scale bars, 600 μm. sc, sclerites embedded in the cuticle; bw, body wall. (*d*) Anterior body in a ventral view with vestibulum (v). (*e*) Posterior body in a vental view with sclerite-free area where the anus (a) and gonopore (go) are situated (reduced mantle cavity). (*f*–*h*) Histological sections of holotype. (*f*) Anterior body with the pharynx ( ph) and anterior midgut caecum (mg). Scale bar, 500 μm. (*g*) Posterior body in the region anterior to the pericardium, with branched seminal receptacles (rs). Scale bar, 200 μm. (*h*) Hindgut region showing the rectum (re) and paired spawning ducts (spd) surrounded by glands (gl). Scale bar, 200 μm. cu, cuticle; de, detritus; gon, gonad; gpd, gonopericardioduct; mg, midgut; pc, pericardium; pd, pericardioduct; sc, sclerite. (Online version in colour.)

The following descriptions are based on the holotype, an adult specimen with an approximate length of 55 mm and a maximum body diameter of 4 mm. Epidermis 50–60 μm thick, lacking glandular cells or papillae. Spines secreted by single enlarged epidermal cells, which elongate into the cuticle and lift the bases of spines. Cuticle up to 250 μm thick. Animal generally uniform in thickness, but close to anterior and posterior body ends, the ventral cuticle is thinner. Epidermal sclerites are solid, flattened spines up to 800 μm long and up to $70 \times 30$ μm at the base. Epidermis underlain by thick layers of circular and longitudinal musculature (figure 2*f*).

Figure 3 shows reconstructions of the anterior and posterior body regions of the holotype based on histology. The mouth opening is located in a small vestibulum, which bears a few papillae (folds). Foregut epithelium high and glandular (figure 2*f*). No foregut glands were observed. A radula is lacking. Pharynx slightly longer than the maximum height of the anterior body. Pharynx with muscular sheet and posteriorly constricted by strong circular musculature; narrow opening between the pharynx and midgut. Midgut wide and uniform, lined by large digestive and glandular cells (figure 2*f*). Long, paired, anteriodorsal caecum and a short anterioventral caecum present. Midgut filling most of the long tubular body and, near the posterior body end, it narrowing to a short ciliated rectum. Anus posterior to the genital opening and surrounded by an area covered in thin cuticle and short sclerites. Remarkably, no mantle cavity is present.

The dorsal paired gonad is well developed, holding both oocytes and spermatocytes. The gonad reaching to the anterior body end, dorsally to the midgut caecum; the median gonad walls fused (figure 2*f*). Pericardioducts (figure 2*g*) short and paired; distinctly ciliated. They fuse just anterior to the relatively narrow pericardium (figure 2*h*), which contains a large, muscular heart ventricle. Short pericardioducts connecting to voluminous spawning ducts that run posteriorly and fuse with each other ventrally to the rectum. Paired seminal receptacles consisting of long and slender ciliated ducts, which anteriorly branch into a number of chambers (figure 2*g*). Both paired and fused parts of the spawning duct lined with ciliated epithelium and surrounded by a thick coat of extraepithelial gland cells, the cell bodies of which come to lie outside a strong

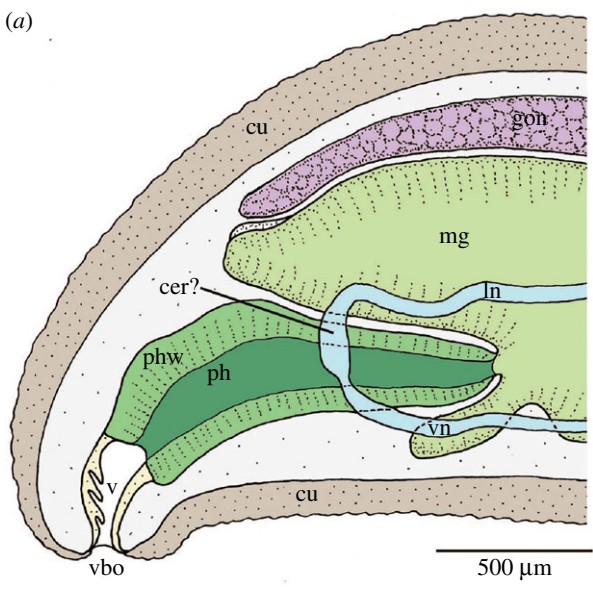

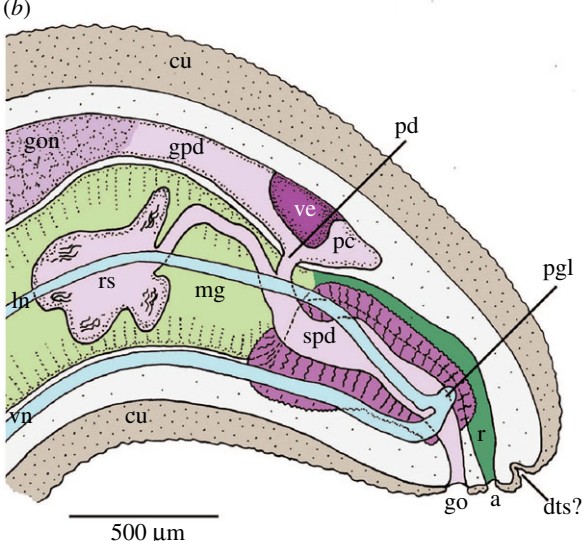

**Figure 3.** Lateral reconstruction of the internal anatomy of the holotype of *A. enigmatica* sp. nov. (*a*) Anterior body. The anteriormost part of the nervous system (cer?) was ambiguous. (*b*) Posterior body. A depression in the cuticle dorsally to the anus could be interpreted as a dorsoterminal sense organ (dts?), but this is doubtful. a, anus; cu, cuticle; gpd, gonopericardioduct; go, gonopore; gon, gonade; ln, lateral nerve cord; mg, midgut; pc, pericardium; pd, pericardioduct; pgl, pedal gland; ph, pharynx; phw, pharynx wall; r, rectum; rs, seminal receptacle; spd, spawning duct; v, vestibulum; vbo, vestibular opening; ve, heart ventricle; vn, ventral nerve cord. (Online version in colour.)

circular muscle layer (figure 2*h*). Single gonopore situated just in front of anal opening and surrounded by an area covered in thin cuticle and small sclerites.

Four major nerve cords run through the entire body, a ventral and a lateral pair. At the posterior end, the ventral and lateral chords of each side are joined by connectives. Reconstruction of the anterior nervous system (including a cerebral ganglion) was not possible based on the two section series available.

## 4. Discussion

Here, we present a phylogenetic framework for Aplacophora that differs dramatically from the current taxonomy of the group and describe a remarkable new solenogaster lacking most of the characters traditionally used to diagnose

Mollusca, significantly expanding known morphological variation in Aplacophora. The unusual morphology of *Apodomenia* initially led us to hypothesize that it represents a 'missing link' between Solenogastres and Caudofoveata. However, all analyses firmly place this species well within Solenogastres. Thus, our results indicate that the foot was secondarily lost at least twice in aplacophoran evolution. Some Palaeozoic chiton-like animals lacking a foot [37–40] have been hypothesized to be stem-group caudofoveates [66,67]. As these animals had chiton-like shells, this hypothesis would suggest independent loss of shells in Caudofoveata and Solenogastres. Although we agree that the available evidence support a chiton-like ancestor for Aplacophora [20,68], independent reduction in the foot in Caudofoveata and *Apodomenia* raises the possibility that Palaeozoic chiton-like taxa without a foot represent additional independent losses. In addition to lacking a foot, *A. enigmatica* is without a mantle cavity or radula, making it among the most extreme deviations from the 'hypothetical archetypical mollusc' [69] known. *Apodomenia enigmatica* sp. nov. demonstrates the striking plasticity of the aplacophoran body plan despite the superficially uniform (worm-shaped) appearance of many members of the group.

Within Solenogastres, we show that several traditionally recognized higher-level taxa (e.g. Aplotegmentaria, Pachytegmentaria and Cavibelonia) are not monophyletic. Cavibelonia was originally defined by the presence of hollow, acicular sclerites [28]. However, some cavibelonians have a scleritome combining scales with hollow acicular sclerites (e.g. Acanthomeniidae) and other species have solid, flattened sclerites (e.g. *Helicoradomenia* spp. [70]). Other characters used in solenogaster taxonomy, such as the radula and ventrolateral foregut glands, are quite variable among taxa ascribed to Cavibelonia. Thus, recovering this clade as polyphyletic was not shocking. Notably, even Salvini-Plawen, who erected the group, expressed his doubts about its validity [8]. Our results are consistent with either multiple independent origins of hollow sclerites (in Amphimeniidae, Pruvotinidae and the last common ancestor of the Epimeniidae/Rhopalomeniidae/Pruvotinidae/Simrothielidae clade) as hypothesized by Salvini-Plawen [8] or multiple independent losses of hollow sclerites (in Neomeniamorpha, Pholidoskepia, Sterrofustia and *Apodomenia*). All solenogasters have solid scales (at least along the foot and around the dorsoterminal sensory organ, if present) and, at least in *Epimenia* and *Proneomenia*, solid scales cover the body of postlarval animals and are later replaced by hollow sclerites [48,71]. We hypothesize that hollow acicular sclerites were present in the last common ancestor of Solenogastres and were modified independently in pholidoskepians, whose scale-like sclerites were likely selected for as an adaptation to a meiofaunal lifestyle, neomeniids, whose harpoon-shaped sclerites appear to grow via a slight modification of the developmental program that produces hollow sclerites in cavibelonians.

Smith *et al.* [17] sequenced an unidentified species of Solenogastres from Greenland. We recollected this species from the same locality and identified it by histology as a pruvotinid (ZMBN 129506–129508). Pruvotinidae was recovered as the sister taxon to the one sampled representative of Sterrofustia, *Phyllomenia*. Sterrofustia is distinguished from the cavibelonian family Pruvotinidae exclusively by the presence of solid sclerites. Pruvotinidae is otherwise a large,

diverse group with species that span a wide range of morphological variation (summarized by García-Álvarez & Salvini-Plawen [24]). Notably, the meiofaunal *Hypomenia sanjuanensis* exhibits a continuum of sclerites with internal cavities ranging in size from those with a cavity that fills around half the volume of the sclerite to those with no hollow cavity at all [49]. Thus, we view the status of Sterrofustia as an order within Solenogastres to be questionable.

Pholidoskepia has been viewed as the extant lineage of Solenogastres with the most plesiomorphic morphological characters [5]. This, combined with the hypothesis that Solenogastres is the sister group to all other Mollusca (e.g. [14]), which is now generally rejected, may have prompted the hypothesis that the last common ancestor of Mollusca was a small, pholidoskepian-like animal [23]. Our results placing large-bodied taxa throughout Solenogastres and Pholidoskepia on a relatively long branch are more consistent with recent work in suggesting the last common ancestor was a relatively large-bodied, chiton-like animal [16] and that the mostly small-bodied Pholidoskepia are relatively derived [72].

We sampled four of the six currently recognized families of Pholidoskepia, and recovered the group monophyletic with strong support. All relationships were strongly supported in BI but placement of Meiomeniidae and Macellomeniidae were weakly supported in ML. Gymnomeniidae has been thought to be closely related to Meiomeniidae as the two families are distinguished almost exclusively on the basis of body size and the number of different sclerite types present [24]. Characters shared by these two taxa include the pedal commissure sac (a unique statocyst-like, geotactic sense organ), an almost complete lack of a basal lamina in the epidermis and a very thin cuticle together resulting in a very fragile integument, ventrolateral foregut glands lacking ducts and the persistence of protonephridia in postlarval or even adult animals [73]. Interestingly, a pedal commissure sac has recently also been found in a meiofaunal dondersiid species [74]. Strong support for a clade of *Macellomenia* and Dondersiidae from BI makes sense in the light of morphology (e.g. same radula type in both families). Sampling of additional members of Pholidoskepia will hopefully help to resolve this issue in the future.

Our results may also shed light on earlier discussions on the plesiomorphic radula type of solenogasters, aplacophorans and molluscs in general. Eernisse & Kerth [75] and Scheltema *et al.* [76] suggested a bipartite (distichous) radula with a medially split radula membrane and two radula teeth or plates in each row to represent the ancestral state. This viewpoint was based on preliminary results on the fossil *Wiwaxia corrugata* and on ontogenetic data for selected chiton and gastropod species. Scheltema [77] later included new fossil findings into her updated interpretation and suggested that a unipartite radula (radula membrane not medially split) with an unpaired central rhachidian tooth and several teeth per row most probably represents the plesiomorphic state for Mollusca. The lack of a rhachidian tooth in aplacophorans is thus interpreted as a derived character. Most interestingly, several early-branching solenogaster clades in our trees do have a unipartite radula, where the single tooth could be homologous to a rhachidian tooth. This includes Amphimeniidae as well as Dondersiidae and Macellomeniidae within Pholidoskepia. Members of Proneomeniidae also have a monopartite radula, but with numerous teeth attached to the radular membrane (polystichous radula). This radula type appears most similar to the radula of other molluscs with a rasping radula, but there seems to be some variation concerning the presence of an unpaired central tooth. Considering the placement of Proneomeniidae, the polystichous radula is unlikely to be a plesiomorphy for Solenogastres. Complete radula reduction can be found not only in *Apodomenia* sp. nov., but in various groups, including all Neomeniidae and many Dondersiidae.

Within Caudofoveata, Limifossoridae exhibits a putatively plesiomorphic distichous radula and a simple body shape [25–27]. However, our results place Prochaetodermatidae sister to Limifossoridae + Chaetodermatidae, consistent with recent studies [44,45]. Prochaetodermatids are small, mostly deep-sea aplacophorans that differ from other caudofoveates by having a paired oral shield, a pair of cuticular jaws, and a radula with two lateral teeth and an undivided radular membrane with a central plate. Interestingly, the long branches separating Prochaetodermatidae and Chaetodermatidae + Limifossoridae show substantial genetic divergence between the two clades. Our results also confirm earlier results indicating that *Chaetoderma* is nested within *Falcidens* [43,44].

## 5. Conclusion

Our results have significantly altered understanding of the evolutionary history and morphological diversity of Aplacophora. Molecular phylogenetics practically turns upside-down previous hypotheses of phylogenetic relationships in both Solenogastres (a large-bodied cavibelonian taxon as the sister group to all other solenogasters) and Caudofoveata (Prochaetodermatidae and not Limifossoridae as sister to all other caudofoveates). Especially in Solenogastres, our results are consistent with a shift from support for the Testaria hypothesis (small-sized pholidoskepian taxa display the most ancestral morphology within Mollusca [22]) to the Aculifera hypothesis (ancestral molluscs were relatively large-bodied, polyplacophoran-like animals [4]). Consequently, evolution of recent aplacophoran molluscs appears to have included several steps of reduction in morphological characters, including the shell(s), digestive gland, broad rasping radula and kidney. Even more extreme reduction is observed in the anomalous *Apodomenia*, which lacks all major characters usually used to define Mollusca.

In addition to advancing understanding of aplacophoran phylogeny, we have dramatically expanded on the previously limited amount of molecular sequence data from aculiferan molluscs by producing deeply sequenced, high-quality Illumina transcriptomes. Our hope is that these data will be of use to researchers addressing a wide variety of questions. We are optimistic that future studies with improved taxon sampling of key lineages not sampled herein (e.g. Acanthomeniidae) will continue to provide insight into the phylogeny and evolution of Aplacophora and Aculifera, thereby shedding more light on the early evolution of Mollusca as a whole.

Competing interests. We declare we have no competing interests.

Funding. This study was supported in part by awards from the US National Science Foundation to K.M.K. (DEB-1210518 and DBI-1306538) and K.M.H. (DEB-1036537, OCE-1155188, ANT-1043745) and an award from the Norwegian Taxonomy Initiative (project no. 70184222) to C.T.

Data accessibility. Data available as part of the electronic supplementary material.

**Acknowledgements.** We thank Akiko Okusu for sharing specimens of *Epimenia babai* and Trish Morse for assistance with collecting meiofaunal solenogasters from Friday Harbor and sharing specimens. We thank the crew and scientists of BioSkag II aboard R/V *Håkon Mosby*, the crew of R/V *Hans Brattström* (University of Bergen), the crew and scientists of the Icy Inverts cruises aboard R/V *Lawrence M. Gould* and *Nathaniel B. Palmer*, the crew and scientists of the IceAGE cruises aboard R/V *Meteor* and R/V *Poseidon*, the crew and scientists of the BoWLs cruises aboard R/V *Oceanus*, the crew and volunteers of the University of Washington Friday Harbor Laboratories R/V *Centennial,* and the FHL administration and staff for supporting specimen collection. We thank Elena Gerasimova who trained KMK in histology and helped with the histological sectioning of specimens. We thank Andrea Kohn and Pam Brannock for help with RNA extraction. We thank M. Carmen Cobo Llovo, Maddie McCutcheon, Rebecca Varney and Meghan Yap-Chiongco for helpful comments on an earlier version of this manuscript. Finally, we thank our departed colleagues Christoffer Schander, Amélie Scheltema and Luitfried von Salvini-Plawen for helpful discussions and inspiration that led to this work. This is Auburn University Marine Biology Program contribution no. 186 and Molette Lab contribution no. 90.

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
