## [Reviewer comments · Proceedings of the Royal Society B: Biological Sciences]

Review History

RSPB-2019-0115.R0 (Original submission)

Review form: Reviewer 1 (Gonzalo Giribet)

Recommendation

Accept with minor revision (please list in comments)

Scientific importance: Is the manuscript an original and important contribution to its field?

Excellent

General interest: Is the paper of sufficient general interest?

Good

Quality of the paper: Is the overall quality of the paper suitable?

Good

Is the length of the paper justified?

Yes

Should the paper be seen by a specialist statistical reviewer?

No

Do you have any concerns about statistical analyses in this paper? If so, please specify them explicitly in your report.

No

It is a condition of publication that authors make their supporting data, code and materials available - either as supplementary material or hosted in an external repository. Please rate, if applicable, the supporting data on the following criteria.

Is it accessible?

Yes

Is it clear?

Yes

Is it adequate?

Yes

Do you have any ethical concerns with this paper?

No

Comments to the Author

This is a most awaited paper on aplacophoran phylogenomics, a clade of molluscs of seminal importance to understand evolution of molluscs, yet poorly understood phylogenetically. The additional description of a species of Solenogastres without foot or radula adds another edge to the paper. The study includes samples collected through many years of hard work and from a diversity of locales (I missed a table with this information and the accession numbers to the used transcriptomes). The methods are state of the art in phylogenomics (the Kocot standard, which is quite different from that of my lab). Given the lack of much conflict (only a couple of nodes are stable, and these are irrelevant to the main points of the paper), there is no need to do much more in terms of analyses, but I wanted to point out that this phylogenomic analysis is quite simple in terms of analyses. Many recent papers include several data matrices and analyze the data under an array of methods that explore additional possible phylogenomic pitfalls. The paper is in general well written (see a couple of specific points below) and should make for a nice article in the Proc. B, as it deals with an entire clade of poorly understood molluscs. My input is just limited to a couple of minor, mostly cosmetic aspects.

The title could be a bit more inspirational...

It took me a while to find out about the size of the two data matrices analyzed. I think that the methods need to include some of this information, which does not appear until the results.

So, there are two data matrices, one with 525 putative orthologs and another one with the 200 least branch length heterogeneous genes, correct? The first matrix was analyzed only under ML (with and without Falcidens caudatus) and the reduced matrix was analyzed under ML and PhyloBayes CAT-GTR. One could also have used IQ-Tree to analyze the larger matrix under a model equivalent to CAT-GTR.

Page 8: Phylobayes should be PhyloBayes

Page 8: “cytochrome c oxidase subunit I” should be “*cytochrome c oxidase subunit I*” [the ‘c’ should be in italics].

Page 9 and onwards: Sorry to pick on this every time but I don’t think it is correct to use the construction “Amphimeniidae was placed sister to all other...”. This or similar formulas are used continuously. I believe that the proper way would be something like “Amphimeniidae was sister group [or sister clade, or sister taxon, or sister family] to all other...”

Is a $pp = 0.68$ “weakly supported”? I think that in Bayesian terms, it is unsupported. (the same clade has a $BS = 55\%$).

“Dondersiidae formed a monophyletic clade...”; a clade is monophyletic by definition, so it should be simply “Dondersiidae formed a clade...” or “Dondersiidae formed a monophyletic group...”

“Falcidens was recovered paraphyletic with respect to Chaetoderma, which was maximally supported.” This sentence is a bit awkward; I assume that what was maximally supported is the sister group between one of the species of Falcidens and Chaetoderma. But also this may be the pace to discuss that *F. caudatus*, the species that clusters with Chaetoderma, has been shown to have an unusual behavior in other analyses? Too bad there were no more Chaetoderma species in these analyses.

In the species description I got a bit confused with the size of the specimens. In one place it says up to 16 cm, in another it says that the largest specimen was 155 mm and the holotype was only 55 mm. The larger specimen was broken at the distal end, but can you estimate how much was missing? Or did you have the two parts of the broken specimen? Is 16 cm the estimate with the missing piece?

The taxonomic section is deficient, as it introduces a new genus that is not described or diagnosed and it is not explicitly said that it is a new genus, and thus it may not be valid under the ICZN. It should also say “gen. nov., sp. nov.” You need to at least add a diagnosis for the genus (then you could say in the species diagnosis “as per the genus diagnosis”), state that it is a new genus (the etymology of the genus and species should not be combined in one section), and you need to identify that this is the type species of the genus by monotypy. Finally, I see that the genus is not assigned to any existing family, so it may also require describing a new family to accommodate it into the current classification system.

Page 13: paleozoic should be Paleozoic.

Page 15: I have to admit that I never knew well the taxonomy of aplacophorans, even though I consider myself to understand mollusc phylogeny pretty well. I thus find it difficult to follow some of the discussions about taxa such as Aplotegmentaria, Pachytegmentaria and Cavibelonia on the fly. I know that the point here is to show that they are not monophyletic, but it may be good to have a figure illustrating the composition and phylogenetic hypothesis of these clades, and also perhaps a proposed classification given the current results.

Page 15: “a least” should be “at least”

Page 17: “Limifossoridae has been viewed as the most plesiomorphic taxon”; I would argue that taxa are not plesiomorphic, but that some of their characters are (while others may be derived).

References: Some titles are all in capitals, some titles are missing italics for genera and species and

some journal titles are in lowercase. This becomes more accentuated towards the end of the references list.

Figure 1: do not italicize "sp." in *Leptochiton* sp.

Figure 2: Can you indicate in the figure legend which specimens (holotype or which paratypes) are illustrated in each image?

A table with the taxonomy of the represented species and accession to SRA numbers should be provided; it should also be indicated which transcriptomes have been generated for this study.

Gonzalo Giribet

Review form: Reviewer 2 (Douglas J. Eernisse)

Recommendation

Accept with minor revision (please list in comments)

Scientific importance: Is the manuscript an original and important contribution to its field?

Excellent

General interest: Is the paper of sufficient general interest?

Excellent

Quality of the paper: Is the overall quality of the paper suitable?

Excellent

Is the length of the paper justified?

Yes

Should the paper be seen by a specialist statistical reviewer?

No

Do you have any concerns about statistical analyses in this paper? If so, please specify them explicitly in your report.

No

It is a condition of publication that authors make their supporting data, code and materials available - either as supplementary material or hosted in an external repository. Please rate, if applicable, the supporting data on the following criteria.

Is it accessible?

Yes

Is it clear?

Yes

Is it adequate?

Yes

Do you have any ethical concerns with this paper?

No

Comments to the Author

This is a splendid manuscript. I am very impressed with the continued quality contributions by this team of aplacophoran workers, and this manuscript represents a major step forward. It contains the best resolved phylogenetic estimate to date of aplacophoran mollusks, which are generally an important but historically problematic taxon for resolving deep molluscan relationships. The phylogenomic analysis is solid. I loved the inclusion of the new species description because of the very bizarre footless attributes of the featured Antarctic solenogaster. I found the authors' review and coverage of the early literature on aplacophoran radular variation (bottom of p. 3, refs 25-27) to be lacking in coverage. Perhaps they missed an opportunity here because, especially because aplacophorans lack shell characters and they claim appropriately that the foot and sclerite characters are more labile than previously recognized, the radula represents an important hard-part feature of interest to those who work on molluscan evolution. Because I have an interest in another aculiferan taxon (chitons) I was very interested in what the improved phylogenetic resolution might reveal about radula variation in aplacophorans. I would respect their opinion if adding discussion of radular variation is beyond the scope of this particular study, but I would love it if they can expand the discussion of it. In our ancient paper (below), we mostly were arguing against an earlier flawed monostichous (not distichous) hypothesis as the plesiomorphic molluscan radular state of both aplacophorans and mollusks, but rereading our paper I see that we included discussion of some of the issues considered in the present paper, and because the authors have demonstrated that some of the aplacophorans with oddball radulas are nested phylogenetically within broad aplacophoran clades with normal radulas, it is now possible to use outgroup comparison to polarize the corresponding states as derived. I would love to see more discussion of this type given that historically the literature on radula variation in aplacophorans is rather confusing. Eernisse, D. J. and K. Kerth. 1988. The initial stages of radular formation in chitons (Mollusca: Polyplacophora). *Malacologia* 28: 95-103. Also, on p. 7 can the authors please better define "with the "-f a" flag settings of RAxML? Douglas J. Eernisse

Decision letter (RSPB-2019-0115.R0)

20-Feb-2019

Dear Mr Kocot:

Your manuscript has now been peer reviewed and the reviews have been assessed by an Associate Editor. The reviewers' comments (not including confidential comments to the Editor) and the comments from the Associate Editor are included at the end of this email for your reference. As you will see, the reviewers and the Associate Editor are positive but have raised some issues with your manuscript that we would like you to address.

Research ethics:

Use of animals and field studies:

Please submit a copy of your revised paper within three weeks. If we do not hear from you within this time your manuscript will be rejected. If you are unable to meet this deadline please let us know as soon as possible, as we may be able to grant a short extension.

Best wishes,
Proceedings B
mailto: proceedingsb@royalsociety.org

Associate Editor
Board Member: 1
Comments to Author:

I agree with the 2 referees, this is an excellent paper. The manuscript would improve with the edits suggested by Prof Giribet. I would also like to suggest them to add a brief discussion on the radula diversity, as suggested by Prof Eernisse, if they have access to that data.

Reviewer(s)' Comments to Author:

Referee: 1

Comments to the Author(s)

This is a most awaited paper on aplacophoran phylogenomics, a clade of molluscs of seminal importance to understand evolution of molluscs, yet poorly understood phylogenetically. The additional description of a species of Solenogastres without foot or radula adds another edge to the paper. The study includes samples collected through many years of hard work and from a diversity of locales (I missed a table with this information and the accession numbers to the used transcriptomes). The methods are state of the art in phylogenomics (the Kocot standard, which is quite different from that of my lab). Given the lack of much conflict (only a couple of nodes are stable, and these are irrelevant to the main points of the paper), there is no need to do much more in terms of analyses, but I wanted to point out that this phylogenomic analysis is quite simple in terms of analyses. Many recent papers include several data matrices and analyze the data under an array of methods that explore additional possible phylogenomic pitfalls. The paper is in general well written (see a couple of specific points below) and should make for a nice article in

the Proc. B, as it deals with an entire clade of poorly understood molluscs. My input is just limited to a couple of minor, mostly cosmetic aspects.

The title could be a bit more inspirational...

It took me a while to find out about the size of the two data matrices analyzed. I think that the methods need to include some of this information, which does not appear until the results.

So, there are two data matrices, one with 525 putative orthologs and another one with the 200 least branch length heterogeneous genes, correct? The first matrix was analyzed only under ML (with and without *Falcidens caudatus*) and the reduced matrix was analyzed under ML and PhyloBayes CAT-GTR. One could also have used IQ-Tree to analyze the larger matrix under a model equivalent to CAT-GTR.

Page 8: Phylobayes should be PhyloBayes

Page 8: "cytochrome c oxidase subunit I" should be "cytochrome c oxidase subunit I" [the 'c' should be in italics].

Page 9 and onwards: Sorry to pick on this every time but I don't think it is correct to use the construction "Amphimeniidae was placed sister to all other...". This or similar formulas are used continuously. I believe that the proper way would be something like "Amphimeniidae was sister group [or sister clade, or sister taxon, or sister family] to all other..."

Is a $pp = 0.68$ "weakly supported"? I think that in Bayesian terms, it is unsupported. (the same clade has a $BS = 55\%$).

"Dondersiidae formed a monophyletic clade..."; a clade is monophyletic by definition, so it should be simply "Dondersiidae formed a clade..." or "Dondersiidae formed a monophyletic group..."

"Falcidens was recovered paraphyletic with respect to Chaetoderma, which was maximally supported." This sentence is a bit awkward; I assume that what was maximally supported is the sister group between one of the species of Falcidens and Chaetoderma. But also this may be the pace to discuss that *F. caudatus*, the species that clusters with Chaetoderma, has been shown to have an unusual behavior in other analyses? Too bad there were no more Chaetoderma species in these analyses.

In the species description I got a bit confused with the size of the specimens. In one place it says up to 16 cm, in another it says that the largest specimen was 155 mm and the holotype was only 55 mm. The larger specimen was broken at the distal end, but can you estimate how much was missing? Or did you have the two parts of the broken specimen? Is 16 cm the estimate with the missing piece?

The taxonomic section is deficient, as it introduces a new genus that is not described or diagnosed and it is not explicitly said that it is a new genus, and thus it may not be valid under the ICZN. It should also say "gen. nov., sp. nov." You need to at least add a diagnosis for the genus (then you could say in the species diagnosis "as per the genus diagnosis"), state that it is a new genus (the etymology of the genus and species should not be combined in one section), and you need to identify that this is the type species of the genus by monotypy. Finally, I see that the genus is not assigned to any existing family, so it may also require describing a new family to accommodate it into the current classification system.

Page 13: paleozoic should be Paleozoic.

Page 15: I have to admit that I never knew well the taxonomy of aplacophorans, even though I consider myself to understand mollusc phylogeny pretty well. I thus find it difficult to follow some of the discussions about taxa such as *Aplotegmentaria*, *Pachytegmentaria* and *Cavibelonia* on the fly. I know that the point here is to show that they are not monophyletic, but it may be good to have a figure illustrating the composition and phylogenetic hypothesis of these clades, and also perhaps a proposed classification given the current results.

Page 15: "a least" should be "at least"

Page 17: "Limifossoridae has been viewed as the most plesiomorphic taxon"; I would argue that taxa are not plesiomorphic, but that some of their characters are (while others may be derived). References: Some titles are all in capitals, some titles are missing italics for genera and species and some journal titles are in lowercase. This becomes more accentuated towards the end of the references list.

Figure 1: do not italicize "sp." in *Leptochiton* sp.

Figure 2: Can you indicate in the figure legend which specimens (holotype or which paratypes) are illustrated in each image?

A table with the taxonomy of the represented species and accession to SRA numbers should be provided; it should also be indicated which transcriptomes have been generated for this study.

Gonzalo Giribet

Referee: 2

Comments to the Author(s)

This is a splendid manuscript. I am very impressed with the continued quality contributions by this team of aplacophoran workers, and this manuscript represents a major step forward. It contains the best resolved phylogenetic estimate to date of aplacophoran mollusks, which are generally an important but historically problematic taxon for resolving deep molluscan relationships. The phylogenomic analysis is solid. I loved the inclusion of the new species description because of the very bizarre footless attributes of the featured Antarctic solenogaster. I found the authors' review and coverage of the early literature on aplacophoran radular variation (bottom of p. 3, refs 25-27) to be lacking in coverage. Perhaps they missed an opportunity here because, especially because aplacophorans lack shell characters and they claim appropriately that the foot and sclerite characters are more labile than previously recognized, the radula represents an important hard-part feature of interest to those who work on molluscan evolution. Because I have an interest in another aculiferan taxon (chitons) I was very interested in what the improved phylogenetic resolution might reveal about radular variation in aplacophorans. I would respect their opinion if adding discussion of radular variation is beyond the scope of this particular study, but I would love it if they can expand the discussion of it. In our ancient paper (below), we mostly were arguing against an earlier flawed monostichous (not distichous) hypothesis as the plesiomorphic molluscan radular state of both aplacophorans and mollusks, but rereading our paper I see that we included discussion of some of the issues considered in the present paper, and because the authors have demonstrated that some of the aplacophorans with oddball radulas are nested phylogenetically within broad aplacophoran clades with normal radulas, it is now possible to use outgroup comparison to polarize the

corresponding states as derived. I would love to see more discussion of this type given that historically the literature on radula variation in aplacophorans is rather confusing.

Eernisse, D. J. and K. Kerth. 1988. The initial stages of radular formation in chitons (Mollusca: Polyplacophora). *Malacologia* 28: 95-103.

Also, on p. 7 can the authors please better define "with the "-f a" flag settings of RAxML?
Douglas J. Eernisse

Author's Response to Decision Letter for (RSPB-2019-0115.R0)

See Appendix A.

Decision letter (RSPB-2019-0115.R1)

18-Apr-2019

Dear Mr Kocot

I am pleased to inform you that your manuscript entitled "Phylogenomics of Aplacophora (Mollusca, Aculifera) and a solenogaster without a foot" has been accepted for publication in *Proceedings B*.

Open Access

Paper charges

Sincerely,

Proceedings B

Associate Editor:

Board Member

Comments to Author:

The two referees (and myself) were very excited with this manuscript. I think the authors have addressed all the major concerns, which were mostly aesthetics. I think it should be published as it is. I would like to congratulate the authors for this great work and their efforts to address the editorial concerns.

Appendix A

Associate Editor

Board Member: 1

Comments to Author:

I agree with the 2 referees, this is an excellent paper. The manuscript would improve with the edits suggested by Prof Giribet. I would also like to suggest them to add a brief discussion on the radula diversity, as suggested by Prof Eernisse, if they have access to that data.

Thank you for the positive feedback. The comments from the reviewers are responded to below in blue text and virtually all of the recommended changes have been made.

Reviewer(s)' Comments to Author:

Referee: 1

Comments to the Author(s)

This is a most awaited paper on aplacophoran phylogenomics, a clade of molluscs of seminal importance to understand evolution of molluscs, yet poorly understood phylogenetically. The additional description of a species of Solenogastres without foot or radula adds another edge to the paper. The study includes samples collected through many years of hard work and from a diversity of locales (I missed a table with this information and the accession numbers to the used transcriptomes). The methods are state of the art in phylogenomics (the Kocot standard, which is quite different from that of my lab). Given the lack of much conflict (only a couple of nodes are stable, and these are irrelevant to the main points of the paper), there is no need to do much more in terms of analyses, but I wanted to point out that this phylogenomic analysis is quite simple in terms of analyses. Many recent papers include several data matrices and analyze the data under an array of methods that explore additional possible phylogenomic pitfalls. The paper is in general well written (see a couple of specific points below) and should make for a nice article in the Proc. B, as it deals with an entire clade of poorly understood molluscs. My input is just limited to a couple of minor, mostly cosmetic aspects.

Thank you for the positive feedback. Specimen collection data are presented in Supplementary Table 1. In terms of the complexity of analyses, we did initially assemble and analyze a large number of datasets to explore factors such as missing data, compositional heterogeneity, saturation, etc. (see Kocot et al. 2017 Syst. Biol. 66(2):256-282), but this shed little light on potential sources of incongruence; the nodes that are weakly supported in the analyses presented here were consistently weakly supported across our other analyses. Because of the strict limitation on word count, we opted to focus more on organismal biology rather than phylogenomic methodology.

The title could be a bit more inspirational...

The title is meant to be a partial parody of the title of this study: Sutton, M. D., & Sigwart, J. D. (2012). A chiton without a foot. *Palaeontology*, 55(2), 401-411.

It took me a while to find out about the size of the two data matrices analyzed. I think that the methods need to include some of this information, which does not appear until the results.

Agreed. This information has been added to the relevant part of the methods.

So, there are two data matrices, one with 525 putative orthologs and another one with the 200 least branch length heterogeneous genes, correct? The first matrix was analyzed only under ML (with and without *Falcidens caudatus*) and the reduced matrix was analyzed under ML and PhyloBayes CAT-GTR. One could also have used IQ-Tree to analyze the larger matrix under a model equivalent to CAT-GTR.

We analyzed all three data matrices with IQ-TREE using the PMSF model. Results are presented in Figure 1 and Supplementary Figures 3, 5, and 7.

Page 8: Phylobayes should be PhyloBayes
Corrected.

Page 8: “cytochrome c oxidase subunit I” should be “cytochrome c oxidase subunit I” [the ‘c’ should be in italics].
Corrected.

Page 9 and onwards: Sorry to pick on this every time but I don’t think it is correct to use the construction “Amphimeniidae was placed sister to all other...”. This or similar formulas are used continuously. I believe that the proper way would be something like “Amphimeniidae was sister group [or sister clade, or sister taxon, or sister family] to all other...”

In some places in the Results section, our wording is a bit Spartan because the manuscript was just under the maximum length permitted by the journal. We have rephrased some of these sentences but our reporting of the results remains brief.

Is a $pp = 0.68$ “weakly supported”? I think that in Bayesian terms, it is unsupported. (the same clade has a $BS = 55\%$).

Agreed. This sentence has been cut.

“Dondersiidae formed a monophyletic clade...”; a clade is monophyletic by definition, so it should be simply “Dondersiidae formed a clade...” or “Dondersiidae formed a monophyletic group...”
Corrected.

“Falcidens was recovered paraphyletic with respect to Chaetoderma, which was maximally supported.” This sentence is a bit awkward; I assume that what was maximally supported is the sister group between one of the species of Falcidens and Chaetoderma. But also this may be the place to discuss that *F. caudatus*, the species that clusters with Chaetoderma, has been shown to have an unusual behavior in other analyses? Too bad there were no more Chaetoderma species in these analyses.

We agree that this was ambiguous. This sentence in the results section has been revised.

In the species description I got a bit confused with the size of the specimens. In one place it says up to 16 cm, in another it says that the largest specimen was 155 mm and the holotype was only 55 mm. The larger specimen was broken at the distal end, but can you estimate how much was missing? Or did you have the two parts of the broken specimen? Is 16 cm the estimate with the missing piece?

The word “about” has been inserted before “16 cm” in the diagnosis to clarify that this is an approximation of the maximum size of the species based on the material collected. The description provides the exact measurement of the largest specimen (paratype 5), which was 155 mm.

The taxonomic section is deficient, as it introduces a new genus that is not described or diagnosed and it is not explicitly said that it is a new genus, and thus it may not be valid under the ICZN. It should also say “gen. nov., sp. nov.” You need to at least add a diagnosis for the genus (then you could say in the species diagnosis “as per the genus diagnosis”), state that it is a new genus (the etymology of the genus and species should not be combined in one section), and you need to identify that this is the type species of the genus by monotypy. Finally, I see that the genus is not assigned to any existing family, so it may also require describing a new family to accommodate it into the current classification system.

Thank you for pointing this out. These omissions have all been corrected.

Page 13: paleozoic should be Paleozoic.
Corrected.

Page 15: I have to admit that I never knew well the taxonomy of aplacophorans, even though I consider myself to understand mollusc phylogeny pretty well. I thus find it difficult to follow some of the discussions about taxa such as Aplotegmentaria, Pachytegumentaria and Cavibelonia on the fly. I know that the point here is to show that they are not monophyletic, but it may be good to have a figure illustrating the composition and phylogenetic hypothesis of these clades, and also perhaps a proposed classification given the current results.

We considered this but the length restriction of the journal does not permit any additional figures. The cited García-Álvarez and Salvini-Plawen 2007 review explains these concepts well and this has been emphasized in the text.

Page 15: “a least” should be “at least”
Corrected.

Page 17: “Limifossoridae has been viewed as the most plesiomorphic taxon”; I would argue that taxa are not plesiomorphic, but that some of their characters are (while others may be derived).
Agreed. This text has been revised.

References: Some titles are all in capitals, some titles are missing italics for genera and species and some journal titles are in lowercase. This becomes more accentuated towards the end of the references list.
The reference list has been carefully proofed.

Figure 1: do not italicize “sp.” in Leptochiton sp.
Corrected.

Figure 2: Can you indicate in the figure legend which specimens (holotype or which paratypes) are illustrated in each image?
Corrected.

A table with the taxonomy of the represented species and accession to SRA numbers should be provided; it should also be indicated which transcriptomes have been generated for this study.
This information is presented in Supplementary Tables 1-2. Transcriptomes generated for this study are now indicated in blue text.

Gonzalo Giribet

Referee: 2

Comments to the Author(s)

This is a splendid manuscript. I am very impressed with the continued quality contributions by this team of aplacophoran workers, and this manuscript represents a major step forward. It contains the best resolved phylogenetic estimate to date of aplacophoran mollusks, which are generally an important but historically problematic taxon for resolving deep molluscan relationships. The phylogenomic analysis is

solid. I loved the inclusion of the new species description because of the very bizarre footless attributes of the featured Antarctic solenogaster.

I found the authors' review and coverage of the early literature on aplacophoran radular variation (bottom of p. 3, refs 25-27) to be lacking in coverage. Perhaps they missed an opportunity here because, especially because aplacophorans lack shell characters and they claim appropriately that the foot and sclerite characters are more labile than previously recognized, the radula represents an important hard-part feature of interest to those who work on molluscan evolution. Because I have an interest in another aculiferan taxon (chitons) I was very interested in what the improved phylogenetic resolution might reveal about radula variation in aplacophorans. I would respect their opinion if adding discussion of radular variation is beyond the scope of this particular study, but I would love it if they can expand the discussion of it. In our ancient paper (below), we mostly were arguing against an earlier flawed monostichous (not distichous) hypothesis as the plesiomorphic molluscan radular state of both aplacophorans and mollusks, but rereading our paper I see that we included discussion of some of the issues considered in the present paper, and because the authors have demonstrated that some of the aplacophorans with oddball radulas are nested phylogenetically within broad aplacophoran clades with normal radulas, it is now possible to use outgroup comparison to polarize the corresponding states as derived. I would love to see more discussion of this type given that historically the literature on radula variation in aplacophorans is rather confusing.

Eernisse, D. J. and K. Kerth. 1988. The initial stages of radular formation in chitons (Mollusca: Polyplacophora). *Malacologia* 28: 95-103.

Thank you for the positive feedback. We have expanded the discussion to address the issue of radular evolution.

Also, on p. 7 can the authors please better define "with the "-f a" flag settings of RAxML?

This sentence now reads "Phylogenetic analyses were conducted for all data matrices using maximum likelihood (ML) in RAxML 7.3.8 [61] with the "-f a" flag, which specifies a search for best-scoring ML tree and a rapid bootstrap analysis and in one program run."

Douglas J. Eernisse